# A No-Brainer! The Therapeutic Potential of TRIM Proteins in Viral and Central Nervous System Diseases

**DOI:** 10.3390/v17040562

**Published:** 2025-04-14

**Authors:** Adam Hage, Mikhaila Janes, Sonja M. Best

**Affiliations:** Innate Immunity and Pathogenesis Section, Laboratory of Neurological Infections and Immunity, Division of Intramural Research, Rocky Mountain Laboratories, National Institute of Allergy and Infectious Diseases, National Institutes of Health, Hamilton, MT 59840, USA; mikhaila.janes@nih.gov (M.J.); sbest@niaid.nih.gov (S.M.B.)

**Keywords:** ubiquitin, tripartite motif (TRIM), E3 ubiquitin ligase, PROTAC, protein degradation, CNS diseases, trim-away, innate immunity, virus infection, disease therapeutic

## Abstract

Tripartite motif (TRIM) proteins comprise an important class of E3 ubiquitin ligases that regulate numerous biological processes including protein expression, cellular signaling pathways, and innate immunity. This ubiquitous participation in fundamental aspects of biology has made TRIM proteins a focus of study in many fields and has illuminated the negative impact they exert when functioning improperly. Disruption of TRIM function has been linked to the success of various pathogens and separately to the occurrence and development of several neurodegenerative diseases, making TRIM proteins an appealing candidate to study for novel therapeutic approaches. Here, we review the current findings on TRIM proteins that demonstrate their analogous properties in the distinct fields of viral infection and central nervous system (CNS) disorders. We also examine recent advancements in drug development and targeted protein degradation as potential strategies for TRIM-mediated therapeutic treatments and discuss the implications these technologies have on future research directions.

## 1. Introduction

The proper functioning of eukaryotic biology is dictated by the ability to rapidly activate and terminate signal transductions. These signaling cascades enable cellular responses to environmental stimuli by controlling fundamental processes in gene expression, cellular differentiation and proliferation, cell survival or apoptosis, and viral restriction and immune responses [1]. To regulate the myriad of events that take place within the cell, post-translational modifications (PTMs) have evolved to rapidly switch these pathways on or off through the attachment or removal of secondary molecules including phosphates, acetyl groups, and ubiquitin (Ub) [2,3]. The importance of PTMs cannot be understated, as entire classes of proteins have evolved to facilitate this role.

The tripartite motif (TRIM) family is a group of E3 Ub ligases consisting of over 80 members in the human genome. TRIM proteins have been in the research limelight for more than 20 years since the discovery of TRIM5α as an HIV-1 restriction factor and TRIM25 in RIG-I-mediated antiviral innate immunity [4,5]. The number of TRIM proteins that have since been identified regulating various cellular pathways has expanded greatly and captivated the interest of researchers from diverse fields. Members of the TRIM family are identified by their conserved N-terminal tripartite RBCC domains, which include the Really Interesting New Gene (RING), one or two B-boxes, and the coiled-coil [6]. The RING domain functions as an E3 Ub ligase by coordinating its two zinc ions to transfer Ub onto a target protein [7]. The B-box and coiled-coil domains promote dimerization and oligomerization, which are required for the E3 ligase activity of some TRIM proteins [7,8,9]. The C-terminus of TRIM proteins is more variable and includes domains like NHL, MATH, and the PRY-SPRY, the latter of which is most common and only found in vertebrates [10]. This diversity allows for a high degree of specificity when targeting substrates. The expansion and diversification of TRIM genes has occurred throughout vertebrate evolution, suggesting positive evolutionary pressure is driving TRIM diversity to regulate increasingly complex biological systems [10,11].

TRIM proteins regulate several biological processes important for health and disease, including cytokine responses, autophagy, innate and adaptive immunity, autoimmune disorders, and cancer [12,13,14,15,16]. Numerous TRIM proteins directly enhance innate immune signaling effectors and are important for the development and function of immune cells [12,13,15]. TRIM proteins can regulate autophagy induction and cargo recognition through ubiquitin-dependent and independent mechanisms and also serve as assembly sites for autophagic machinery [14,17]. Furthermore, nearly one-third of TRIM proteins have been associated with cancer development either through the enhancement of oncogenic potential or tumor suppressor loss-of-function [1,16].

As a consequence of their fundamental involvement in so many cellular processes, dysregulation of TRIM proteins can be detrimental to homeostasis. Loss of TRIM function has been linked as a contributing factor to a variety of pathological conditions, including oncogenesis, impaired innate immune responses, and onset of central nervous system (CNS) diseases. The growing appreciation for TRIM function has also garnered them increasing consideration as targets for drug discovery. An emerging method for treating diseases caused by viruses or pathogenic proteins has been to engage the ubiquitin proteasome system (UPS) to induce targeted protein degradation (TPD). By bioengineering TRIM proteins to employ the cellular degradation machinery against viral and CNS diseases, TPD strategies like proteolysis-targeting chimeras (PROTACs), monovalent molecular glues (MGs), or Trim-Away have gained traction as novel therapeutic approaches [18].

The properties that allow TRIM proteins to recognize and resolve viral threats parallel with how they respond to proteinopathies and neurodegeneration. Aggregated proteins that seed neurological disorders are themselves a form of molecular pattern. Sensing these misfolded proteins and redirecting them for degradation illustrates a consensus mechanism used by TRIM proteins in two very different scenarios. While many TRIM-based therapeutic applications began as ways to counter viruses, these strategies are now heavily utilized in neurobiology research. In this review, we provide an overview on the current TRIM literature linking their involvement to infectious and neurological diseases. We also highlight the approaches and technologies being used to establish TRIM proteins as practical therapeutic options. Although we focus on the participation of select TRIM proteins in infectious and CNS disease, other excellent TRIM reviews spotlighting their broad impacts are also available [2,19,20,21,22,23].

## 2. Trim Proteins: Targets for Viral Disease

The importance of TRIM proteins in host antiviral defense is unmistakable as the field has grown exponentially in recent years. The steady pace of publications highlighting how TRIM proteins orchestrate pathogen removal through antagonism of the viral replication cycle or modulation of host signaling pathways has cemented TRIM research as a major arena for understanding host defenses. However, a surge in genome-wide screening has increasingly found TRIM proteins to be usurped by viruses to fill replication machinery roles that these pathogens otherwise lack. These discoveries paint an unorthodox picture of TRIM–virus interactions as a double-edged sword. In the wrong hands, these viral degraders can be used to greatly expand a particular viruses’ toolbox and therefore warrant serious consideration for therapeutic targeting.

### 2.1. TRIM23

RNAi screening of TRIM proteins identified TRIM23 as essential for autophagy in response to viral infection [24]. Loss of TRIM23 impaired autophagy induced by several viruses including herpes simplex virus-1 (HSV-1), influenza A virus (IAV), and encephalomyocarditis virus. Viral infection promotes TRIM23 autoubiquitination through K27-linked polyubiquitin on its ARF domain which activates an additional TRIM23 GTPase function. Hydrolysis of GTP promotes TRIM23-TBK1 associations, TBK1 dimerization and autophosphorylation, and ultimately TBK1 activity. This cascade phosphorylates p62, an autophagy receptor, allowing it to recognize and remove viral components.

Some of the first indications that canonically antiviral TRIM proteins may moonlight as direct promoters of viral replication came from a genome-wide CRISPR/Cas9 screen used to identify pro-viral IAV host factors [25]. Identification of TRIM23 as a pro-IAV host factor in the initial screens of this study was further validated with *TRIM23* knockout cell lines. Loss of TRIM23 impaired replication of multiple strains of IAV (H1N1, H3N2, and H5N1) as well as vesicular stomatitis virus, suggesting TRIM23 may be required for sustaining replication in a broadly proviral manner [25]. Surprisingly, these data are at odds with the described role for TRIM23 in promoting IAV-induced autophagy and suggest the functional capabilities of TRIM proteins differ in response to specific viral pathogens. It is possible that TRIM23 promotes IAV replication through an unrecognized mechanism that is independent of its autophagy-mediated restriction for other viruses.

Prior to this work, TRIM23 was found to be a critical inhibitor of type-I interferon (IFN-I) signaling by the NS5 protein of yellow fever virus (YFV) [26]. Ubiquitination of the viral NS5 protein is enhanced in the presence of IFN-I, leading to the promotion of its subsequent antagonism functions. This finding prompted the authors to investigate whether TRIM proteins were the source of this Ub due to their importance in modulating IFN-I signaling. Overexpression screening revealed TRIM23 as responsible for NS5 ubiquitination and knockdowns of TRIM23 significantly impaired this PTM event and ensuing viral replication [26].

### 2.2. TRIM6

TRIM6 enhances IFN-I signaling through activation of IKKε [27]. By synthesizing unanchored K48-linked polyubiquitin chains, TRIM6 promotes IKKε oligomerization and autophosphorylation, leading to an optimal antiviral state both in vitro and in vivo. While the E3 Ub ligase activity of TRIM6 is required for proper formation of IFN-I signaling complexes against IAV and West Nile virus (WNV) [27,28,29], this function has been usurped by Ebola virus (EBOV) to promote its own replication [30]. CRISPR-mediated knockout of TRIM6 significantly hampered EBOV replication. This loss in viral titer was caused by failure of the EBOV VP35 protein to be ubiquitinated on residue K309, leading to reduced function of VP35 in its role as a cofactor for the viral polymerase. Mutation of VP35 K309 to prevent Ub conjugation, or use of the catalytically defective TRIM6 C15A RING mutant, was sufficient for disruption of EBOV replication as measured by a minigenome assay [30]. These phenomena were further validated using recombinant EBOV mutants where the lysine at residue 309 was exchanged for either a glycine or arginine (K309G or K309R) to prevent Ub conjugation [31]. Loss of Ub at this residue resulted in severe replication deficiencies of EBOV when grown in IFN-I competent and incompetent cell lines, indicating that the block occurs independently of innate immune activation. Mechanistically, TRIM6-derived Ub enhanced EBOV transcriptase function by promoting interactions between VP35 and the viral polymerase while limiting associations with nucleoprotein, thereby preventing premature packaging of nucleocapsids into nascent virions [31]. This specificity of TRIM6 function during various viral infections showcases how TRIM proteins can have roles in host defense against one pathogen while becoming a target to directly promote the replication of another. These studies call attention to the innovative concept of TRIM proteins directly influencing pathogen success and therein identify opportunities for pharmacological countermeasure development.

### 2.3. TRIM7

TRIM7 cellular functions include regulation of tumor invasiveness, apoptosis, and antiviral IFN-I responses [32,33,34]. Notably, TRIM7 restricts multiple human enteroviruses through ubiquitination and proteasomal degradation of the viral 2BC protein, a non-structural protein that plays a crucial role in viral replication and membrane remodeling, thereby inhibiting viral RNA replication [35]. Although TRIM7 can hamper viruses through degradation of viral proteins, it has also been shown to promote virus replication by facilitating viral entry [36,37]. Mass spectrometry analysis of flavivirus-infected cells revealed the presence of a conserved di-glycine motif characteristic of Ub PTMs on the envelope protein of several flaviviruses, including dengue virus, WNV, YFV, and Zika virus (ZIKV) [36]. A recombinant ZIKV bearing a mutation at this residue preventing ubiquitination (lysine to arginine at position 38 ‘K38R’) showed a marked reduction in replication for both in vitro and in vivo studies. Mice infected with the K38R mutant ZIKV displayed enhanced survival and lower viral titers in several tissues including the brain [36]. TRIM7 was hypothesized as the likely E3 ligase responsible for envelope ubiquitination from an earlier genome-wide siRNA knockdown screen identifying TRIM7 as a proviral host factor for YFV and from its known expression in tissues relevant for flavivirus infection like the brain [38,39]. Indeed, both siRNA-mediated knockdown and CRISPR knockout of TRIM7 impaired replication of wild-type ZIKV to levels matching the K38R mutant. The generation of TRIM7-null mice confirmed TRIM7 was required for optimal ZIKV replication in permissive tissues including the brain, reproductive tissues, and eyes [36]. TRIM7 enhanced ZIKV replication by conjugating Ub to the viral envelope during infection, enabling budding virions to better attach to host receptors for efficient cellular entry [36]. While it is currently unknown where in the cell TRIM7-mediated ubiquitination of the ZIKV envelope occurs, it is hypothesized this takes place in and around the intracellular membranes of the Golgi compartment. Confocal microscopy and subcellular fractionation identified TRIM7 relocation to the Golgi during ZIKV infection where it colocalized with the viral envelope [36].

### 2.4. TRIM28 and TRIM33

TRIM28 and TRIM33 hold documented roles as antiviral proteins; TRIM28 promotes IFN-I signaling through K63-linked ubiquitination of TBK1 while TRIM33 targets the HIV-1 integrase for proteasomal degradation [40,41]. However, both proteins were identified in a genome-wide CRISPR/Cas9 screen promoting replication of severe acute respiratory syndrome coronavirus 2 (SARS-CoV-2) [42]. Additionally, exploration of the SARS-CoV-2 nucleocapsid interactome using tandem mass spectrometry identified multiple host factors involved in protein SUMOylation as likely binding partners including TRIM28 [43]. SUMOylation was confirmed to be critical for the SARS-CoV-2 nucleocapsid to efficiently bind viral RNA and form liquid–liquid phase separation (LLPS) droplets. LLPS inclusions are ideal for assembling viral proteins, nucleic acids, and cellular factors in a makeshift viral factory that protect the virus from host innate immune sensors. TRIM28 co-precipitated with SARS-CoV-2 nucleocapsid and promoted its SUMOylation while the enzymatically inactive C651A TRIM28 could not. Knockdown of TRIM28 impaired nucleocapsid oligomerization, association with RNA, and LLPS formation. Prevention of this TRIM28-SARS-CoV-2 nucleocapsid interaction and LLPS formation with synthesized interfering peptides significantly impeded SUMOylation, reduced SARS-CoV-2 titers, and rescued IFN-I responses [43].

### 2.5. TRIM5α

In the twenty years since its first description as an HIV-1 restriction factor [4], TRIM5α has had a prominent impact on the field’s understanding of how TRIM structure and function contributes to viral restriction. Multiple domains of TRIM5α are involved in interactions with the HIV-1 capsid [9,44,45]. By forming a complementary hexagonal lattice structure around the viral capsid, TRIM5α promotes HIV-1 clearance through premature capsid disassembly and activation of innate immunity [46]. For much of its study, TRIM5α has only been described as a retroviral restriction factor. Only recently has TRIM5α gained appreciation for sensing other viral families and targeting their components for degradation. TRIM5α from both human and rhesus macaques potently restricts tick-borne orthoflavivirus replication [47]. Mapping of TRIM5α interactions revealed the viral NS2B/3 protease to be the target for K48-linked polyubiquitination and proteasomal degradation. Furthermore, TRIM5α has now been found to restrict DNA viruses including the orthopoxvirus vaccinia virus. TRIM5α recognition of the viral L3 capsid protein limits viral replication and induces innate immunity [48].

While all studies of TRIM5α thus far have described its antiviral functions, the possibility that TRIM5α functions in a proviral manner has been suggested. A study interested in mapping host genetic variations that drive fatal Ebola virus disease (EVD) utilized the Collaborative Cross recombinant inbred mouse population and identified the *Trim5* locus as a susceptibility loci for EVD [49]. Deletion of the *Trim5* locus on chromosome 7 removed the murine *Trim5* orthologs (*Trim12a*, *Trim12c*, *Trim30a*, *Trim30b*, *Trim30c*, and *Trim30d*), leading to improved survival and less severe EVD after Ebola virus infection compared to wild-type littermates. RNA-Seq analysis determined the *Trim5* locus-null animals showed a milder inflammatory response while animals that retained the *Trim5* locus displayed elevated markers typical of cytokine storm. This lack of exuberant inflammatory responses correlated with prolonged survival and a reduction in genes associated with severe liver disease [49]. Murine *Trim12c* positively enhances IFN-I and NFκB activation through ubiquitination of TRAF6 and may contribute to the increased host susceptibility to EVD [50]. A direct proviral role for TRIM5α has not yet been described and the function of murine *Trim5* orthologs is not equatable to human TRIM5α. However, these studies suggest Ub-mediated innate immune activation by TRIM5α is potentially detrimental for diseases that exacerbate inflammation [46].

## 3. Therapeutic Trim Proteins in Neurodegenerative Disease

Many hallmark CNS diseases including Alzheimer’s disease, Parkinson’s disease, and Huntington’s disease are characterized by intracellular neurofibrillary tangles (NFTs) comprised of hyperphosphorylated insoluble tau proteins [51]. Collectively defined as tauopathies, the development of, and inability to resolve, these abnormal protein assemblies are a uniting neuropathological feature for the more than 20 associated CNS disorders [52]. Protein quality control (PQC) systems are essential to maintain protein homeostasis [53]. A properly functioning PQC system is essential for cell survival and overall well-being. Failures in PQC operation can have far-reaching effects on neuronal homeostasis and are associated with neurodegenerative diseases [54]. By leveraging a network of molecular chaperones, PQC systems intercept misfolded and aggregated proteins to efficiently target defective and redundant proteins for elimination or recycling [55]. PQC systems also comprise protein disaggregases which resolve established deposits of faulty proteins as a second layer of protection [56]. The mechanisms of disaggregation are a burgeoning field in neurodegenerative research as PQC function has been shown to diminish with age, contributing to the onset of several CNS diseases. TRIM proteins are central to PQC pathways, with recent investigations revealing their participation in the prevention and resolution of tauopathies [57]. These findings expand the utility of TRIM proteins in eukaryote biology and present an exciting new frontier in TRIM research.

### 3.1. Alzheimer’s Disease

Alzheimer’s disease (AD) is one of the most prevalent causes of dementia, contributing to over half of all global cases [58]. AD is a secondary tauopathy, distinguished by the added presence of extracellular amyloid β (Aβ) plaques in addition to NFTs comprised of hyperphosphorylated tau [51]. The presence of NFTs and Aβ plaques is strongly correlated with cognitive impairment in neurodegenerative diseases like AD, making their formation and resolution a primary research focus for investigators [59].

To systematically approach the question of TRIM involvement in CNS disease, Zhang and colleagues cloned 75 human TRIM proteins into expression vectors and tested their ability to degrade a GFP-tagged tau construct bearing the P301L familial tauopathy mutation known to promote aggregate formation [56,58]. Expression of TRIM10, 11, 26, 36, and 55 reduced tau aggregates by 50–100% [56]. Complementing this overexpression screening, both siRNA-mediated knockdown and CRISPR knockout strategies were employed to determine if TRIM loss conversely promoted tau aggregation. Strikingly, TRIM11 produced the strongest outcomes in each assay by degrading GFP-P301L entirely when overexpressed or by increasing GFP-P301L levels by 2–3-fold when endogenous TRIM11 expression was reduced.

Importantly, TRIM11 has been linked to the onset of progressive supranuclear palsy (PSP), a prevalent form of sporadic tauopathy, wherein *TRIM11* single nucleotide polymorphisms are associated with disease [60]. Supporting a general role for TRIM11 in human tauopathies, brain samples from 23 sporadic AD brains showed a marked reduction in TRIM11 protein expression when compared to 14 age- and sex-matched healthy controls. This absence in TRIM11 was confirmed at the individual neuron level and suggests the loss of TRIM11 may have a direct role in promoting AD (Figure 1A). Mechanistically, the authors determined TRIM11 functions as both a molecular chaperone and disaggregase for tau by directly binding and promoting the SUMOylation of tau for proteasome-mediated degradation and by maintaining solubility of tau to prevent self-association (Figure 1B–D) [56].

The biological relevance of TRIM11 was demonstrated experimentally using murine-derived primary neurons as well as transgenic mouse models for AD. Delivery of TRIM11 to the CNS via the adeno-associated virus AAV9 vector protected 3×Tg-AD mice, a mouse model resembling human AD, by reducing levels of hyperphosphorylated tau and limiting cognitive and behavioral deficits [56]. The multifaceted ways TRIM11 prevents tauopathies suggests TRIM proteins may be valuable countermeasures to treat CNS disorders. While the authors limited the latter portion of their investigation to TRIM11, it would be of interest to discern whether the other potent TRIM proteins identified in the screen (TRIM10 and TRIM55) possess similar, possibly redundant, roles in protecting neuronal tissues. In support of this possibility, highly cooperative and multi-layered TRIM defenses have been observed in the context of virus infection. For example, TRIM14, 22, and 41 all target the nucleoprotein of IAV to the proteasome for degradation [2,61,62,63].

### 3.2. Parkinson’s Disease

Parkinson’s disease (PD) is the second most common form of neurodegenerative disease distinguished by the presence of Lewy bodies (LBs) and Lewy neurites (LNs), comprising amyloid fibrils of α-synuclein (α-Syn) [64]. While the development of an accurate early diagnostic test remains challenging [65], contemporary studies using cohort screenings have yielded a growing list of genetic subtypes and variants associated with a risk of developing PD [66]. Alterations in the transcription of several TRIM genes from PD-derived iPSCs suggest they may have unrecognized roles in neurodegenerative processes related to PD [67]. Furthermore, whole exome sequencing examining early-onset PD in a cohort of 743 patients revealed variants in 13 TRIM proteins as potential risk markers for PD, including previously implicated members TRIM11 and TRIM9 [68,69].

TRIM11 has garnered recent appreciation, particularly in neuronal tissues, as having a central role in PQC by enhancing the degradation of aberrant and normal proteins and augmenting the overall rate of proteolysis [70]. One of the earliest indications TRIM11 resolves CNS diseases was described using a mouse model of PD [71]. Zhu and colleagues first identified TRIM11 suppressing de novo α-Syn fibrilization, increasing the solubility of α-Syn, and preventing the formation of mature α-Syn fibrils [71]. Strikingly, only a very small amount of TRIM11 was required to observe these effects, with TRIM11 resolving protein aggregates even when faced with a 140-fold excess of α-Syn substrate [71]. Using both real-time quaking-induced conversion (RT-QuIC) and luciferase-based aggregate resolution assays, the authors sophistically showed TRIM11 dissolves pre-formed fibrils of α-Syn and restores the enzymatic activity of malformed luciferase trapped in aggregates, respectively [71].

Remarkably, TRIM11 can identify the fold state of a protein and distinguish misfolded proteins from normally folded aggregates. While TRIM11 normally resides in the cytoplasm, expression of a defective protein linked to familial neurodegenerative diseases (ataxin-1 with an expanded polyglutamine region: Atxn1 82Q) induced the relocalization of TRIM11 to the nucleus where it bound Atxn1 82Q. This movement of TRIM11 was specifically promoted by the pathogenic state of the protein, as it did not colocalize with the normal Atxn1 30Q form. Importantly, the authors note that TRIM11 chaperone/disaggregase activities are structurally separable from its SUMO ligase activity. Removal of the RING domain prevented TRIM11 SUMOylation, but chaperone/disaggregase activities were retained. While TRIM11 can SUMOylate misfolded proteins for proteasomal clearance, TRIM11 did not SUMOylate α-Syn fibrils. Furthermore, mutations that inactivate TRIM11 SUMO ligase activity did not impair its α-Syn disaggregase activity [71]. Multiple TRIM proteins have been identified as having SUMOylation activity through a screen using Mdm2 as a substrate [72]. Interestingly, while TRIM11 was included in this screen, it was unable to SUMOylate Mdm2 [72]. This preference for SUMOylating aberrant proteins like Atxn1 82Q, but not normal proteins like α-Syn and Mdm2, suggests molecular determinants such as the folding state of a substrate may determine whether a TRIM protein utilizes ubiquitination or SUMOylation.

Delivery of α-Syn pre-formed fibrils intracranially to the dorsal striatum of mice recapitulates the LB/LN-like aggregation and neurodegeneration phenotypes found in human PD [73]. Using this mouse model of PD, AAV9-delivered TRIM11 showed protective effects by limiting α-Syn pathology by 65–80% in the frontal and piriform cortexes. This translated to improved locomotor activity, reduced anxiety, and shorter periods of movement freeze in mice given TRIM11 compared to the GFP control [71]. Collectively, these studies provide a clear picture of TRIM11 involvement in PD by preventing protein aggregation, limiting neurodegeneration, and rescuing motor impairment. The identification of TRIM11 as a critical host factor involved in the resolution of α-Syn aggregates, combined with later studies revealing its role in resolving pathogenic tau fibrils, strongly support a role for PQC components as essential for the maintenance of CNS homeostasis [56,71].

### 3.3. Huntington’s Disease

Huntington’s disease (HD) is an incurable, progressive neurodegenerative disease marked by a combination of motor, cognitive, and behavioral symptoms [74]. HD is caused by an autosomal dominant inherited mutation resulting in an expanded CAG trinucleotide repeat in the huntingtin gene (HTT) [75]. This results in expression of huntingtin with polyglutamine (polyQ) sequences that exceed normal length, which predispose it to fragmentation and aggregation [74]. Similarly to AD and PD, the failure to maintain protein homeostasis is a central component of HD. Unsurprisingly, the participation of TRIM proteins and the UPS in HD has been implicated as both TRIM19 and TRIM37 have been described as playing protective roles [76,77].

TRIM19 was shown to interact with polyQ aggregates associated with HD and several spinocerebellar ataxias (SCAs) [78]. Specifically, TRIM19 promotes the degradation of mutant ataxin-7, the causative protein for SCA type-7 [79]. TRIM19 expression can reduce levels of misfolded Atxn1 82Q as well as aggregates of pathogenic HTT (Httex1p 97QP) by SUMOylation, as the SUMO-resistant form of pathogenic HTT (Httex1p 97QP(KR)) resisted degradation by TRIM19. Ultimately, TRIM19 was found to cooperate with an additional E3 Ub ligase (RNF4) to direct mutant HTT to the proteasome in a step-wise manner requiring both SUMO and Ub [76].

Despite ubiquitous expression of HTT in the brain and surrounding tissues, neurodegeneration observed in HD preferentially occurs in the striatum. In a nonhuman primate (NHP) model of HD [77], HTT degradation transpired at a much slower rate in the NHP striatum and RNA-Seq analysis from these tissues identified a number of downregulated UPS genes including TRIM37. Surprisingly, TRIM37 is capable of binding to HTT from both NHP and human cortical lysates, but not from mice, identifying the interaction between TRIM37 and HTT as potentially primate-specific. TRIM37 expression facilitated the ubiquitination and degradation of HTT while the siRNA-mediated loss of TRIM37 prevented HTT ubiquitination and increased HTT stability. In vivo delivery of AAV9-TRIM37 reduced HTT aggregation while a lentiviral-delivered shRNA-TRIM37 reduced endogenous TRIM37 levels and promoted HTT aggregation in the striatum [77].

## 4. Potential Targeting Strategies for Trim Proteins

The notion of targeting a protein of interest (POI) for destruction via one of the cellular degradation machineries was first outlined in 1991 [80]. Since then, TPD has employed diverse technologies utilizing proteasomal and lysosomal degradation strategies [81]. TPD technologies like Trim-Away, PROTACs, and RING-Bait have the potential to clear disease-causing proteins that are challenging to target or otherwise deemed undruggable by contemporary small-molecule inhibitors, particularly in CNS diseases [82]. The recent period of rapid advancement in the TPD field marks a watershed moment for TRIM research as interest in the utility of these E3 Ub ligases as emerging therapeutics continues to grow.

### 4.1. TRIM21

The first description of TRIM21 as a potent mediator of viral restriction was in 2010 and its utility in antiviral research has been well-documented since [83,84,85,86,87]. Complexing of the PRY-SPRY domains of TRIM21 homodimers with the Fc portion of antibodies transforms TRIM21 into a high-affinity cytosolic antibody receptor [86,88]. This unique ability allows TRIM21 to sense intracellular pathogens that have evaded antibody opsonization with high specificity, setting it apart from the more commonplace direct sensing by other TRIM family members. Similarly to other TRIM proteins, TRIM21 directs its antibody-bound targets to the proteasome for degradation. Interaction between TRIM21 and the antibody–virus complex is a multistep process involving viral binding to host cell receptors, endocytosis, and lysis of the endosomal membrane. For adenoviruses, interaction with the host cell receptors CAR and αv-β3/5 integrins initiates binding events leading to the loss of the fiber protein, exposure of the membrane lytic protein VI, and subsequent lysis of the endosomal membrane [89,90]. Recognition of internalized Fc domains triggers TRIM21 auto-ubiquitination on its RING domain, which redirects TRIM21 and its antibody-bound cargo to the UPS. The removal of invading pathogens by these means has the added benefit of activating downstream antiviral innate immune signaling pathways. Recycling of the bound Ub chains post-TRIM21 degradation allows subsequent activation of required components for IFN-I and proinflammatory cytokine production [91]. Leveraging these highly conserved IgG Fc interactions allows TRIM21 to provide an added line of defense against a great number of diverse pathogens.

### 4.2. Trim-Away

In 2017, the James and Schuh labs first described “Trim-Away” as a novel method for directly altering endogenous protein levels using TRIM21 [92]. By repurposing its cytosolic antibody receptor function, TRIM21 can be engineered to bind the Fc portion of an antibody targeting a protein of interest, enabling TRIM21-mediated ubiquitination and degradation of the antibody-bound target (Figure 2A). Mirroring how TRIM21 aids the antiviral immune response by targeting pathogens bound by high-specificity antibodies [86], TRIM21 removal of GFP was only attainable when anti-GFP antibody, and not the IgG control, was used to facilitate its capture [92]. The utility of Trim-Away was supported by its ability to locate and remove targets from different regions within the cell. Membrane-anchored GFP, chromatin-incorporated GFP, and GFP sequestered in the nucleus by a nuclear localization signal were all rapidly removed by the combination of anti-GFP antibody and TRIM21. Importantly, these studies provided proof-of-concept that Trim-Away is a viable strategy for the treatment of CNS disorders like Huntington’s disease [93]. Trim-Away can be directed to pathogenic forms of proteins, while sparing the healthy versions, provided an antibody specifically recognizing the disease conformation is available. Utilizing the 3B5H10 antibody specific for the disease-causing form of the HTT protein, mutant HTT was cleared while normal HTT levels were preserved [92,94]. This revelation that TRIM21 can be used to remove harmful CNS proteins has been further expanded to encompass pathogenic tau assemblies [95,96]. TRIM21 is expressed in neural cells and is rapidly recruited to remove tau aggregates when anti-tau antibodies have marked them for targeting [95,97]. Recently, in vivo evidence has validated the application of Trim-Away in mouse models of tau pathology. Protection from tau aggregates by TRIM21 in a mouse model was lost in TRIM21-null animals and delivery of Trim-Away components by AAV systems rescued mice from excessive tau pathology [98,99]. Trim-Away harnesses cellular PQC systems and leverages the availability of conformation-specific reagents to target host and viral proteomes. It is therefore possible that Trim-Away systems can one day be deployed to efficiently target a wide spectrum of native cellular proteins in a rapid and specific manner.

### 4.3. RING-Bait

The ability to selectively recognize and degrade multimeric proteins has gained TRIM21 traction as a promising way to overcome difficulties in removing established intracellular aggregates. Since TRIM21 requires clustering of its RING domain to initiate ubiquitination and degradation [100,101], incorporation of the RING domain into fusion chimeras has allowed for selective targeting of multimerized forms of tau proteins by bringing enough RING domains in contact upon aggregation [99]. While these methods have utilized nanobody technologies as delivery platforms, a new approach has been to fuse these catalytic drivers of TRIM proteins directly to an offending protein “bait”. The resulting “RING-Bait” can focus the degradative potential of TRIM21 to large disease-causing aggregates without impacting the target’s pool of functional monomers (Figure 2B) [102].

Fusing the RING domain of TRIM21 to the aggregate-forming tau mutant P301S generated a potent bait that eliminated seeded tau aggregates in reporter cells [102]. Doxycycline-inducible RING-Bait molecules with an mCherry reporter showed direct incorporation into assembling tau aggregates, leading to their subsequent removal. RING-Bait, like other TRIM21-based systems, utilized the proteasome to achieve tau removal. Mutations in the RING portion, that prevent ubiquitination by blocking RING dimerization or by rendering dimers catalytically inactive (M72E and I18R respectively), failed to degrade tau aggregates. RING-Bait has showed potential for resolving increasingly relevant models of CNS disease. Seeding of tau aggregates from post-mortem brain extracts of AD and PSP patients was significantly impaired by RING-Bait, highlighting its efficacy at resolving aggregates for different tauopathy fold types. These protective characteristics were not limited to 2D immortalized tissue culture models but were also shown to function in both primary neurons and in vivo. AAV-delivered RING-Bait protected primary neurons from seeded aggregation in neuronal bodies and processes. Injection of AAV-delivered RING-Bait or the M72E and I18R mutant variants into Tg2541 mice expressing P301S tau showed a significant reduction in tau aggregates in the frontal cortex for wild-type, but not the mutant, RING-Bait. These results were reproducible using either tail vein or stereotaxic injection routes and translated to improved motor function in animals as measured by the number of hind leg movements needed to traverse a measured walkway [102]. The RING-Bait system allows for the removal of aberrant aggregates only as they appear through selective incorporation into the developing pathogenic forms, sparing the healthy, soluble versions of the protein. This simplicity lends itself to applications for other pertinent CNS diseases and raises the question of whether other TRIM proteins may be engineered to a similar degree.

### 4.4. PROTACs

First described in 2001, PROTAC protein degraders have rapidly moved from academic research to therapeutic development by major pharmaceutical companies [103,104]. PROTACs are heterobifunctional molecules that function as bait to recruit a POI and an E3 Ub ligase. By bringing both substrates in close proximity, the PROTAC allows the target-seeking portion and an E3 ligase to conjugate Ub onto the POI. The now ubiquitinated POI is diverted to the UPS for removal while the PROTAC and E3 ligase are free to repeat this process on another instance of the POI (Figure 2C) [104,105]. Use of the endogenous cellular machinery sustains PROTAC-mediated effects for longer periods with a high-degree of specificity. TRIM E3 ligases may offer PROTACs expanded utility as therapeutics due to their large number of members, tissue-selective expression patterns, and preference for their native substrates [57,82].

A recent example demonstrating proof-of-concept for TRIM-based PROTACs showed the functionality of TRIM21 to alter the course of tumor progression by selectively degrading Human antigen R (HuR) [106]. Elevated levels of HuR are associated with tumorigenesis in multiple cancer types and poor patient prognosis thereby making HuR ideal for focused therapeutic research. Fusion of a single domain antibody (VHH) which binds HuR (VHH^HuR^) to an Fc domain enables TRIM21 to recognize the Fc portion of the VHH^HuR^ PROTAC and rapidly remove endogenous HuR. Preservation of HuR was conferred by introduction of the H433A mutation into the PROTAC Fc domain, which prevents TRIM21 attachment. To overcome the limitation of adequate TRIM21 expression in certain tissues, Fletcher and colleagues created a TRIM21-based ‘bioPROTAC’ by fusing TRIM21 directly to VHH^HuR^. Both transfection and doxycycline-inducible models of the bioPROTAC promoted rapid turnover of HuR via the proteasome, which halted cancer cell proliferation in vitro and arrested tumor growth in vivo [106].

Further adaptation of TRIM21 by replacement of the Fc-binding portion (the PRY-SPRY domain) with the third fibronectin type III domain of human tenascin-C (Tn3) conferred new degradation specificity for TRIM21 [107]. TRIM21-Tn3 hybrids with Tn3 domains that interact with two B-cell lymphoma-associated genes, MALT1 and EED, were able to degrade both the GFP-fusion forms as well as the native versions of both targets [107]. Synthesizing TRIM21 bioPROTACs that do not require an antibody intermediary to precisely target a POI makes it possible to utilize the native E3 ligase repertoire of any tissue to selectively eliminate troublesome proteins.

The potential to eliminate proteins that cause disease has garnered TRIM21 significant attention as a premier TPD platform. Lu and colleagues have recently described two additional modes of TRIM21-mediated TPD: in conjunction with the monovalent MG (S)-ACE-OH to target nuclear pore complex (NPC) proteins and as a PROTAC with specificity for higher-order assembly targets as seen in biomolecular condensates [82]. Monovalent MGs are small molecules which can alter protein–protein interactions by inducing proximity between factors, stabilizing interactions, or by inducing new interactions [108]. Utilizing high-throughput screening of 81,845 small drug-like molecules, the authors determined (S)-ACE-OH acts as a monovalent MG degrader guiding the relocalization of TRIM21 to the NPC where it promotes the degradation of multiple nuclear pore factors [82]. Prior reports have identified clustering of substrates as a defining trigger for TRIM21-based Trim-Away technologies [93]. To address whether this function of TRIM21 is broadly applicable to multimeric assemblies and similar biomolecular condensates, the authors generated two PROTACs (TrimTAC1 and 2) which resolved several biomolecular condensates generated by artificial fusion proteins while ignoring these same constructs in their soluble state (Figure 2C). These abnormal protein condensates comprising membrane-less assemblies enriched in proteins have been suggested to play a role in neurodegeneration [109]. The level of precision described by the above PROTAC studies indicates that the selective depletion of pathogenic, but not soluble, forms of host factors is attainable.

### 4.5. TRIM-Based Antiviral Therapeutics

Although many of the proof-of-concept studies leveraging TRIM proteins as therapeutics have been demonstrated using models of CNS disease, there is a growing interest in optimizing TRIM21 platforms to better respond to the heterogeneity of viruses. These new TRIM21-based systems expand the engineering toolbox and increase the list of diseases TRIM proteins can be employed against.

Numerous approaches were explored to combat SARS-CoV-2 in the early years of the pandemic, including TRIM21-based degradation platforms. Specifically, Chatterjee and colleagues used computational methods to design peptides to bind the SARS-CoV-2 spike protein and promote intracellular clearance of the virus through TRIM21 [110]. The peptide was designed in silico using structures of SARS-CoV-2 spike protein bound to soluble ACE2 (the cellular entry receptor) to develop a 23 mer peptide with high-affinity to the spike protein receptor binding domain. This peptide was fused to an antibody Fc domain to allow for TRIM21 recognition. In a simple inhibition assay, pseudotyped lentiviruses expressing the full length SARS-CoV-2 spike protein were less capable of infection in the presence of the interfering peptide [110]. Screening available crystal structures for in silico-based peptide design could be extended to viruses that infect the CNS and for which therapeutic development is challenging.

The increased interest in Trim-Away technologies has spurred innovation in designing increasingly better antigen–antibody bait systems with nanobodies as one of the most popularized. Derived from camelid heavy-chain-only antibodies, nanobodies are smaller than their conventional IgG counterparts, allowing easier binding to antigens whose epitopes are difficult to recognize [111]. The use of smaller antibody fragments in transient transfection systems overcomes the need for electroporation or microinjection, one of the biggest hurdles of Trim-Away. A recent report put nanobody-based Trim-Away to the test using African swine fever virus (ASFV) as an example [112]. Plasmid constructs encoding for pig TRIM21 and specific nanobodies against the ASFV structural proteins (p30, p54, and p72) were transfected into pig cells infected with ASFV or expressing one of the three structural protein targets. Each of the three TRIM nanobodies reduced levels of their ASFV targets through different degradation pathways. Surprisingly, loss of both p30 and p54 required a functional proteasome while p72 removal was impaired when cells were treated with the lysosomal inhibitor chloroquine. Importantly, all three TRIM21 nanobodies showed effective degradation of their targets during infection of ASFV, leading to a negative impact in viral replication [112].

The use of TRIM proteins as TPD tools against viral infection continues to gain momentum and recent work from Ma, Yin, and colleagues highlights the advantage of the improved accessibility of Trim-Away systems. By developing a series of controllable TPD systems using TRIM21, the authors programed degradation of HSV-1 [113]. Screening truncation variants of TRIM21 revealed that loss of the B-box domain promoted the degradation of a d2EGFP reporter by 90% compared to only 30% observed with full-length TRIM21. To activate Trim-Away using novel methods (chemical or light-based triggers), this truncated “ΔTRIM21” was incorporated into the REDMAP system to initiate ΔTRIM21 transcription upon exposure to 660 nm wavelength red light [114]. Length of exposure and light intensity screening showed an ideal parameter of 12 h at 1000 µW to degrade 88% of the d2EGFP reporter in cells. When d2EGFP was replaced with a monoclonal antibody against the HSV-1 glycoprotein D (gD), this “RedΔTRIM21-TPD” system was now able to target the virus. An engineered EGFP-labeled HSV-1 virus showed a 60–75% reduction in GFP signal after activation of RedΔTRIM21-TPD. Efficient degradation of intracellular HSV-1 was observed for several different multiplicities of infection, making this simple light-based system a promising method for controlled therapy. In addition to RedΔTRIM21-TPD, deployment of the ΔTRIM21 variant was explored using more novel delivery platforms including chemical triggers to control CRISPR-Cas9 gene editing and 440 nm blue light for inhibition of tumor growth in vivo [113]. Even with its more aggressive degradation capabilities, the authors acknowledge ΔTRIM21 showed limited efficacy towards insoluble proteins and large aggregates. Nevertheless, the versatility of programming therapeutic activation with simple light systems or clinically approved drugs makes this technology highly accessible.

## 5. Conclusions and Future Perspectives

The last 20 years of research has greatly expanded our understanding of TRIM proteins as essential regulators of biological processes from cellular function to pathogen invasion. In addition to their well-established role as antiviral restriction factors, TRIM proteins have been the subject of exciting advancements in PQC, the resolution of CNS diseases, and pioneering technologies in novel therapeutics. For a TRIM-based therapeutic to become a reality, significant challenges and unresolved questions will need to be addressed.

It will be essential to determine the negative effects of TRIM function on cellular regulation. For example, while TRIM overexpression screening clearly demonstrated resolution of the soluble and insoluble forms of a P301L tau construct by TRIM11, a large proportion of the overexpressed TRIM proteins appeared to promote expression levels of both soluble and insoluble P301L tau [56]. Furthermore, although TRIM6 can enhance IKKε-mediated IFN-I signaling to establish a protective antiviral response, a recent study identified TRIM6 expression as a driving factor for PD in a mouse model [115]. TRIM6 overexpression promoted higher levels of α-Syn and suppression of TRIM6 expression by the long non-coding RNA LINC02282 limited neuronal damage in PD mice by driving DNA methyltransferases to silence the TRIM6 promoter region. Results like these may temper the expectations for TRIM proteins as therapeutics due to cell-type, and disease-specific requirements that allow them to alleviate one disease state but exacerbate another. Therefore, the development of strategies which can finely tune TRIM function will be critical for therapeutic intervention.

One way to overcome off-target effects may be through tightly controlled delivery methods like stereotaxic injection or cell-type specific AAV9 systems as shown in the context of TRIM11 in the CNS [56]. These strategies may be necessary to avoid altering the innate immune signaling and IFN-I production ability of TRIM11 [116]. In addition, using state-of-the-art concepts like RING-Bait to specifically and exclusively target a pathogenic protein to the catalytic domain of a target-seeking TRIM bypasses the need for intermediates like antibodies or monovalent MGs [102]. The substrate-induced clustering of the TRIM21 RING domain may also be applicable to other TRIM proteins which require multimeric assemblies to activate their functions. By leveraging the large number of TRIM family members and their high-degree of tissue specificity, these technologies may provide a tailored approach for treating a wide array of CNS, virological, and autoimmune diseases. Another possibility would be to leverage the structurally separable Ub/SUMO ligase and chaperone/disaggregase activities of TRIM proteins. TRIM11 requires its RING domain for SUMOylation, but not for its chaperone and disaggregase activities. By using truncated forms of TRIM proteins that only lack Ub/SUMO ligase activity, it may be possible to remove pathogenic protein aggregates without instigating additional signaling pathways. Given that TRIM19 and TRIM21 both prevented and reversed protein aggregation in a similar manner to TRIM11 [71], it will be important to assess whether future TRIM-based therapeutics can be achieved without the use of the UPS.

TRIM research in the field of virology has excelled at developing detailed mechanistic descriptions of how TRIM proteins alter the outcome of disease. The mechanisms of action for both TRIM5α and TRIM25 are comprehensive and include their structural and functional properties, mechanisms of target recognition and restriction, and thorough mapping of signaling pathways. Mechanistic studies including identification of the binding interface between a TRIM and its pathogenic protein target and understanding what are the molecular or structural determinates that regulate whether a TRIM uses ubiquitination or SUMOylation will be important for addressing gaps in our understanding of how TRIM proteins can be therapeutic in CNS diseases.

TRIM proteins have dual roles in both antiviral defense and CNS homeostasis. Interdisciplinary approaches will be needed to integrate TRIM research in virology and neuroscience to aid in the successful development of CNS disorder treatments. The achievements made using TRIM proteins in TPD paint a bright future for the field of TRIM-based therapeutics. It will be exciting to witness the creative solutions the next 20 years of TRIM research brings.

## Figures and Tables

**Figure 1 viruses-17-00562-f001:**
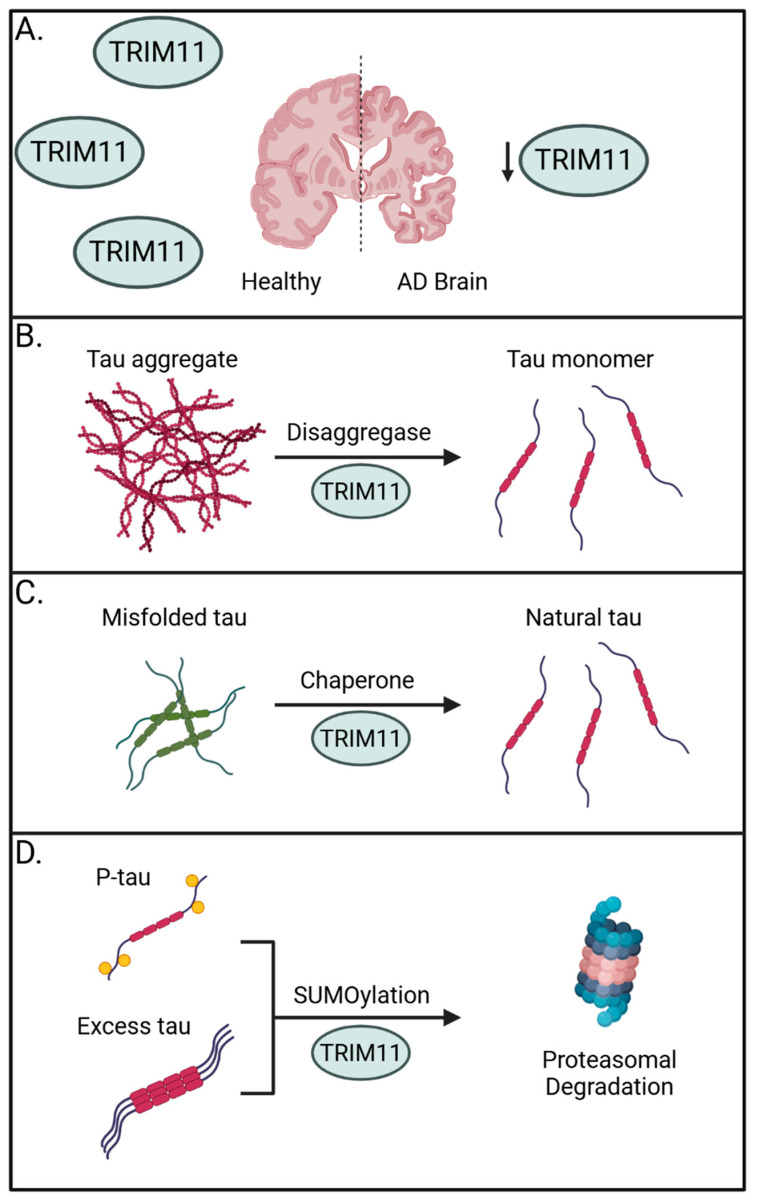
The preventative roles of TRIM11 against tauopathies. (**A**) Sporadic AD brains have reduced expression of TRIM11, suggesting its importance in countering the onset of tauopathies. (**B**) TRIM11 can operate as a disaggregase to promote solubility of tau aggregates and fibrils. (**C**) TRIM11 functions as a molecular chaperone to restore misfolded tau and maintain levels of natural tau. (**D**) TRIM11 SUMOylates hyperphosphorylated and excess tau, marking them for proteasomal degradation. Created with BioRender.com.

**Figure 2 viruses-17-00562-f002:**
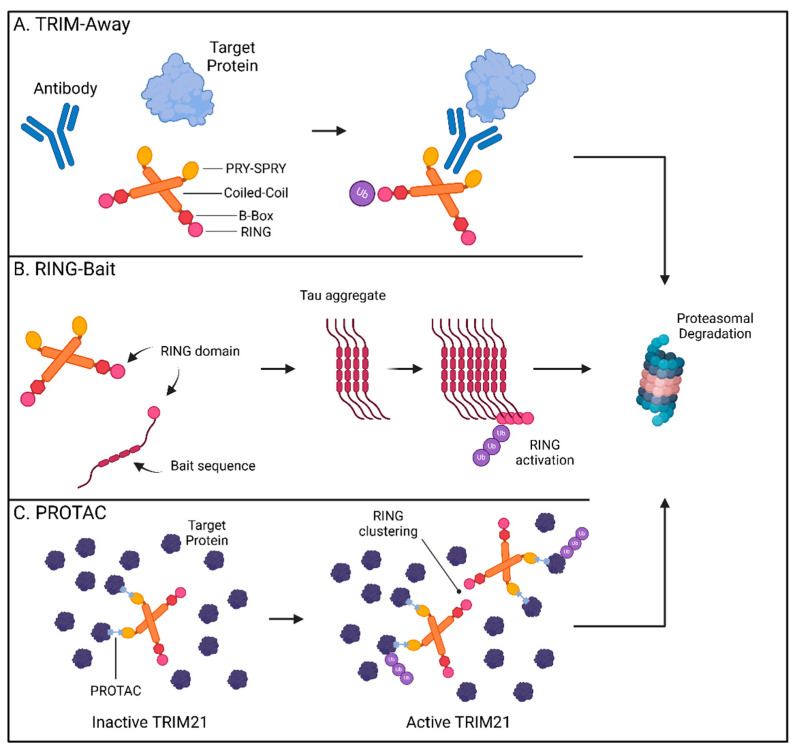
TRIM21-based technologies for targeted protein degradation therapeutics. (**A**) TRIM-Away: TRIM21 binds to the Fc-region of an antibody bound to a target protein promoting the ubiquitination and proteasomal degradation of the antibody and its cargo. (**B**) RING-Bait: The RING domain of TRIM21 can be fused with a bait molecule consisting of an aggregate-prone tau sequence resulting in its incorporation into the assembling tau fibrils. Clustering of the RING domains promotes the ubiquitination and proteasomal degradation of tau aggregates. (**C**) TRIM21-based PROTACs promote interactions between TRIM21 and multimeric proteins. Substrate-induced clustering activates the E3 ubiquitin ligase function of TRIM21 to selectively degrade its multimeric targets thereby preventing protein aggregation. Created with BioRender.com.

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
