# Peer review of "A No-Brainer! The Therapeutic Potential of TRIM Proteins in Viral and Central Nervous System Diseases"

_viruses, 2025, doi:10.3390/v17040562_

Round 1
Reviewer 1 Report
Comments and Suggestions for Authors
The manuscript ‘A No-brainer! The Therapeutic Potential of TRIM Proteins in Viral and Central Nervous System Diseases’ by Hage et al. describes the role of TRIM E3 ligase family proteins in the context of viral infection as well as in neurodegenerative disorders including Alzheimer’s disease, Parkinson’s disease, and Huntington’s disease. The review article highlights that the TRIM proteins are central to protein quality control pathways in addition to their other functions. The authors further discussed the progress in druggable strategies for using TRIM proteins against viral infections and various neurodegenerative disorders.
The manuscript is coherent, with well-articulated descriptions of major discoveries in the field. The figures are well crafted and have clear illustrations. Below are some suggestions to further strengthen the article for the broader readership.
1) The authors nicely introduced TRIM proteins and their potential therapeutic applications. It would be helpful to describe the major mechanisms by which TRIMs generally function in health and disease (e.g., regulation of cytokine responses, autophagy, proteasomal pathway, etc.) in a short paragraph. This would help the readers to better understand the subsequent sections where the authors describe specific TRIMs.
2) The authors noted in lines 37-39 that the discovery of TRIM5α and TRIM25 was a cornerstone in TRIM research evolution. While TRIM5α was covered in detail, it may be helpful to also discuss TRIM25 as one of the most extensively studied TRIM proteins in the context of viral infection. The authors should focus on highlighting novel functions/mechanisms of TRIM25 that were identified recently, such as RNA binding (e.g. PMID: 30342007), TRIM25 condensation in stress granules for activating RIG-I signaling (PMID: 38750080), its regulation by U1 snRNA for RIG-I ubiquitination (PMID: 38483900), and the recent structural insights into RIG-I regulation by TRIM25 (PMID: 40024477). Furthermore, the authors could summarize TRIM25’s emerging role as a ZAP co-factor (e.g. PMID: 28060952 and PMID: 28202764).
3) The authors have described various therapeutic strategies for TRIMs. Certain functions of TRIM proteins are independent of their E3 ligase activity (for example, disaggregate activity of TRIM11), and TRIM functions are not limited to protein degradation. It would be informative to briefly discuss other unexplored avenues for TRIM-based therapeutics against viral infections/CNS pathologies.
4) In the first paragraph (line 33), it would be helpful for the general readers if the authors could define what the “secondary molecules” are.
Reviewer 2 Report
Comments and Suggestions for Authors
In this review by Hage, Janes and Best, the authors discuss the critical and dual roles of TRIM proteins specifically in viral infections and CNS disorders. They also summarize the strategies used in the field to harness TRIM E3 ligase function for targeted protein degradation as therapies.
Major points:
- While the authors provide a valuable summary of the therapeutic potential of TRIM function through strategies like Trim-Away, RING-bait, and PROTAC, the review could be further enriched by incorporating additional comprehensive sources for readers. I suggest referencing other reviews on TRIM proteins and their roles in CNS disorders: Cell Mol Neurobiol (2023 Mar 29;43(6):2567–2589) and Front. Mol. Neurosci. (Volume 16 – 2023, https://doi.org/10.3389/fnmol.2023.1287257.
- Can the authors discuss/clarify the current hypothesis for how the antibody-virus complex access the cytoplasm after endocytosis for TRIM21 interaction?
- Can the authors also discuss/clarify current hypothesis for how TRIM7 ubiquitinates flavivirus envelopes? Is TRIM7 localized to the ER/ERGIC/Golgi?
- The authors could consider adding following to the conclusion and perspectives: a) the requirement for mechanistic studies focused on of how TRIM proteins interact with misfolded proteins and regulate cellular pathways in CNS disorders. b) The authors have also correctly pointed out the beneficial or detrimental nature of TRIM protein function. Hence developing strategies that allow to finely tune TRIM function will be critical for therapeutic interventions. c) The authors have done a good job of showcasing examples of TRIM protein function in virology and neurobiology. Therefore, I feel that a statement exemplifying the importance of interdisciplinary approaches to integrate virology and neuroscience would help to tie things up. This would be necessary to understand the dual roles played by TRIM proteins in antiviral defense and CNS homeostasis for enhancing success of therapeutic interventions.
Minor: The review unfortunately is full of colloquial designations that I urge the authors to avoid in formal scientific writing. Overall, the authors should use a clearer and more scientifically rigorous phrasing to avoid any ambiguity throughout the review. Below, I have listed only a few examples.
1. “TRIM proteins” or “TRIM E3 ligases” instead of “TRIMs”
2. dysregulated, aberrant, or pathogenic instead of “problematic proteins”
3. neurobiology research instead of CNS field.
4. degradation instead of termination.
5. Please avoid personifying of proteins: TRIM25’s, TRIM23’s, TRIM11’s, VP35’s, these “designer TRIM21’s” etc.
6. “This cascade phosphorylates p62, an autophagy receptor” instead of “This cascade phosphorylates the p62 autophagy receptor”.
7. culprit, mousetrap? Please consider using better scientific terms.
8. Line 135: Please provide context to improve comprehension: 2BC protein of enteroviruses is a non-structural protein that plays a crucial role in viral replication and membrane remodeling.
Comments on the Quality of English LanguageThe review unfortunately is full of colloquial designations that I urge the authors to avoid in formal scientific writing. Overall, the authors should use a clearer and more scientifically rigorous phrasing to avoid any ambiguity throughout the review. Please see the comments for details
Reviewer 3 Report
Comments and Suggestions for Authors
The article titled ‘A No-brainer! The Therapeutic Potential of TRIM Proteins in Viral and Central Nervous System Diseases’ by Hage et al. describes the role of the E3 ligase family (TRIM) proteins in a myriad of fundamental cellular processes and their link to various viral infection as well as complex neurodegenerative disorders characterized by neurofibrillary tangles (including Alzheimer’s disease, Parkinson’s disease and Huntington’s disease) potentially caused due to the errors or overloading of the protein quality control systems. TRIM proteins are central to protein quality control pathways, making TRIM proteins unique druggable targets for viral infections and various neurodegenerative disorders.
The manuscript is fluent and coherent, with well-articulated observations and rationales. Here are my few comments to further strengthen the article-
Major points:
- In para 2.1, the authors described siRNA screen demonstrating autophagy-dependent restriction of HSV-1 and CRISPR/Cas9 screen delineating the pro-viral role of TRIM23. The siRNA-mediated screen also showed the involvement of TRIM23 in IAV-induced autophagy. The authors may comment on the probable association of TRIM23 during IAV infection.
- Similarly, the specificity of TRIM6 during various viral infections can also be briefly described.
- TRIM proteins facilitate both ubiquitination and SUMOylation, potentially resulting in proteasomal degradation. The authors can postulate the determinants (molecular determinants or structural) that regulate ubiquitination vs SUMOylation.
- Certain functions of TRIM proteins are independent of their E3 ligase activity. How do authors hypothesize these functions can be utilized for the therapeutic interventions?
- The references can be in the order of the description of details in the statement. For example, the line number 36-39, the references for TRIM5α and TRIM25 can be rearranged.
